# Understanding the Use of Heterogenous Data in Tackling Urban Flooding: An Integrative Literature Review

**Ming Ren** [1] **, Ziqi Zhang** [2,*] **, Jun Zhang** [3] **and Luca Mora** [3,4]

1 School of Information Resource Management, Renmin University of China, Beijing 100086, China; renm@ruc.edu.cn
2 Information School, University of Sheffield, Sheffield S1 4DP, UK
3 The Business School, Edinburgh Napier University, Edinburgh EH11 4BH, UK; o.zhang@napier.ac.uk (J.Z.); l.mora@napier.ac.uk (L.M.)
4 Academy of Architecture and Urban Studies, Tallinn University of Technology, 12616 Tallinn, Estonia
* Correspondence: ziqi.zhang@sheffield.ac.uk

**Abstract:** Data-driven approaches to urban flooding management require a comprehensive understanding of how heterogenous data are leveraged in tackling this problem. In this paper, we conduct an integrative review of related studies, and this is structured based on two angles: tasks and data. From the selected 69 articles on this topic, diverse tasks in tackling urban flooding are identified and categorized into eight categories, and heterogeneous data are summarized by their content type and source into eight categories. The links between tasks and data are identified by synthesizing what data are used to support the tasks in the studies. The task–data links are a many-to-many relationship in the sense that one particular data category supports multiple tasks, and one particular task uses data from multiple categories. The future research opportunities are also discussed based on our observations. This paper serves a signpost for researchers who wish to gain an overview of the heterogenous data and their use in this field and lays a foundation for studies that aim to develop a data-driven approach to tackle urban flooding.

**Keywords:** data; urban waterlogging; urban flooding; integrative literature review

## 1. Introduction

With the acceleration of global warming and rapid urbanization, the last decade has seen an increase in both the frequency and intensity of urban flooding disasters at a global scale, whose effects have severely impacted on economic development and societal stability. In the context of this paper, urban flooding refers to pluvial flooding in urban areas that are typically caused by extreme rainfall, which surpasses the capacity of urban drainage systems. They have become one of the most frequent and serious natural hazards in many cities around the world, especially those that are subject to frequent torrential rainfall, undergoing rapid population growth and urbanization processes, or where urban drainage systems are outdated and the upgrade is lagging.

Emergency managers interpret disasters as recurring events whose control focuses on four phases: mitigation, preparedness, response, and recovery [1]. During the entire cycle, it is critical to share data and knowledge for effective decision making. On the one hand, disasters are usually a continuous and changeable process, with no clear boundaries between the phases. This calls for continuous monitoring and data sharing across the phases [2]. On the other hand, there are multiple individuals and organizations with different backgrounds and expertise, which requires communication and collaboration across different levels and locations [3].

The acquisition, storage, and elaboration of large-scale, multi-modal data has become more affordable due to advancement and diffusion of smart city technologies, such as Internet of Things (IoT) solutions, sensor networks, and cloud computing [4]. Urban flooding

research largely acknowledges that the combination of data and case-based reasoning can provide relevant insight into natural disaster reduction (e.g., [5–8]). Despite these premises, a comprehensive understanding is still missing on how heterogeneous data should be leveraged in urban flooding management.

Most of the existing systems offer singular functions that are designed to satisfy specific user needs, however, may not meet needs of other user communities. One of the reasons is that the related tasks are diverse, and the data required for analysis are highly heterogeneous in form and interdisciplinary and distributed in nature [8]. In addition, the links between the tasks and data are unclear, which makes it difficult to decide on appropriate data to be collected for a specific task.

Data-driven approaches to urban flooding management require the consideration of task–data configurations. Although several studies have acknowledged the importance of such a goal, they dealt with the problem at a conceptual level or used ontology to model the tasks and data but without specifying what data are required and how tasks are associated [2,9]. This paper aims to understand the use of heterogeneous data in tackling urban flooding. Specifically, this paper answers the following research questions.

RQ1. What are the tasks in urban flooding management?

RQ2. What are the data involved in urban flooding management?

RQ3. What data are used to support a specific task?

To this end, we conducted an integrative literature review, which focused on articles that report data-driven approaches for urban flooding management and covered 69 peer-reviewed papers that explain what data are used to accomplish tasks in tackling this problem. The review is structured based on two angles: tasks and data. The diverse tasks are identified and grouped into eight categories, and the objectives and relationships are also clarified.

The heterogeneous data are summarized in different ways, by the content (eight main data categories) and by the source (eight main types of sources). The links between tasks and data are identified by synthesizing what data are used to support the tasks in the studies. This paper serves a signpost for researchers who wish to gain an overview of the research in this domain. It also lays the foundation for studies that aim to develop a data-driven approach to tackle urban flooding, as they can be informed of the types of technologies, systems, and data that need to be considered for integration. We further make recommendations for future research and practice based on our observations.

The remainder of this paper is structured as follows. Section 2 presents the integrative literature review methodology. Section 3 summarizes the tasks in terms of the objectives and the methods (where available). Section 4 summarizes the data in terms of the content type (i.e., data categories) and the sources. The links between tasks and data as well as research opportunities are discussed in Section 5. The paper is concluded in Section 6.

## 2. Methodology

Whilst the use of heterogeneous data has been explored across multiple disciplines to address various types of emergencies, little is known regarding what kinds of data are used to address urban flooding problems, or by what means. An integrative literature review does not simply synthesize representative literature on a topic but adds new insights via an analytical synthesis of literature [10–12], and as such it is considered a useful method to advance current knowledge and advance future research on a specific topic [13]. Following this approach, this study seeks to understand the tasks and data addressed in the previous work on urban flooding emergency management and how data are used to support those tasks.

This integrative review is based on the guidelines proposed by Whittemore and Knafl [14] that illustrate a five-stage strategy for enhancing the rigor of the review process, including problem identification, literature search, data evaluation, data analysis, and data presentation. The process of this integrative review is shown in Figure 1. In the problem identification stage, the purpose of this integrative literature review was defined

as to develop a holistic understanding of the use of heterogeneous data in urban flooding emergency management. Though task diversity and data heterogeneity in this field have been observed, there is a lack of integrative review of the tasks, data, and their links. We did not pursue a complete coverage of all studies of flooding management but rather aimed to develop a comprehensive understanding of heterogeneous data, associated tasks, and their links through the more recent and earlier seminal studies that had an empirical focus.

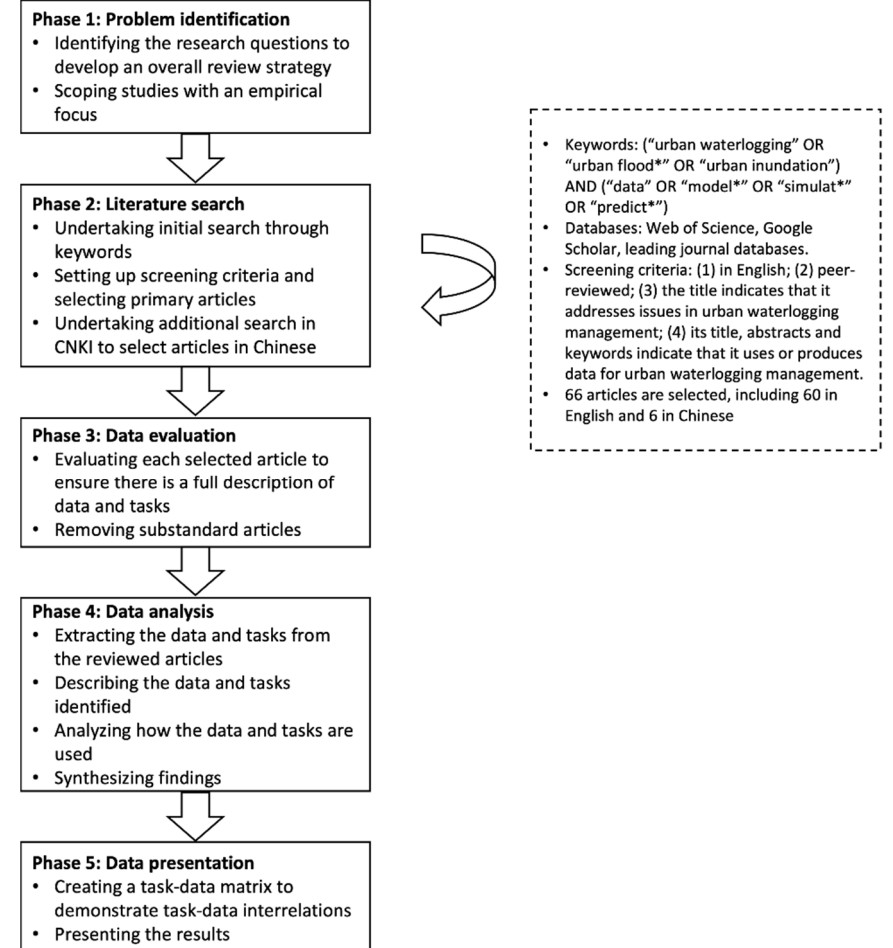

**Figure 1.** Five phases of the integrative literature review.

In the second stage, we searched databases focusing on articles ranging across multiple disciplines using keywords ("urban waterlogging" OR "urban flood*" OR "urban inundation") AND ("data" OR "model*" OR "simulat*" OR "predict*") in Google Scholar (available online at https://scholar.google.com/, accessed on 20 May 2022, Web of Science (available online at https://www.webofscience.com/wos/woscc/basic-search, accessed on 20 May 2022), and in several leading journals on information studies, geography, and computer sciences. The timespan 'since 2010' was used to focus on the recent studies first. The retrieved articles were filtered based on a set of criteria.

An article was included if: (1) it is in English; (2) it is a peer-reviewed publication (e.g., journal articles, conference proceedings); (3) its title indicates that it addresses issues in urban flooding management; and (4) its title, abstracts and keywords indicate that it uses or produces data for urban flooding management. A snowball sampling approach was also applied that the seminal work published before 2010 were also included through the references of filtered articles. The search process continued until we noticed a convergence in terms of the tasks and data covered by the literature.

During the search, we noticed that the research done by Chinese scholars or institutions accounted for more than half of the literature, suggesting that there might be uncovered

findings in the published work in Chinese. Thus, a second round of search was conducted in China National Knowledge Infrastructure (CNKI) (available online at https://en.cnki.com.cn/, accessed on 20 May 2022), focusing on the studies that belong to an under-represented topic by the selected articles in English. In total, six articles in Chinese were selected. Until October 2021, 66 articles were selected in this integrative review, including 60 in English and six in Chinese. In the revision, three articles in English were added into the selection.

In the third stage of data evaluation, each selected article was reviewed to ensure it provided a full description of using or producing certain type of data in dealing with specific tasks in managing urban flooding. The authors split the workload with each reviewing a subset of the selected articles and held regular meetings to discuss the articles they reviewed and how to summarize their findings.

In the fourth stage of data analysis, the authors worked separately to extract the tasks and data from the reviewed articles. We first agreed on a process for categorizing their tasks and data, with each following a coding process independently. Further, a number of focus groups were conducted in order to merge and consolidate all of the codes developed. The description of data and tasks involved in the reviewed studies as well as how they were utilized was also analyzed for relevance.

In the final stage of data presentation, a task–data matrix was developed to synthesize and demonstrate task–data interrelations across multiple sources, which will be shown in following sections.

## 3. Tasks Involved in Urban Flooding Management

Managing urban flooding involves a number of tasks, which are categorized as inundation simulation and prediction, risk analysis, flood monitoring, response and evacuation planning, trend analysis, cause analysis, conceptual modelling, and policy analysis. These tasks are interrelated, for example, cause analysis and risk analysis often rely on the methods of, or results from inundation simulation and prediction.

### 3.1. Inundation Simulation and Prediction

This task focuses on simulating the formation and progression of urban flooding, with a goal to predict the inundation. Since inundation prediction is a complex task, methods need to make simplified assumptions. Typically, an area of interest is divided into a grid-based catchment, and then three main groups of factors are calculated for each grid, i.e., soil runoff (or infiltration), surface flow, and drainage flow. Simulation results vary from binary output of inundation [15,16], to more fine-grained inundation level in centimeters [17–19].

Soil runoff measures the amount of direct runoff from a rainfall event, and depends on parameters, such as rainfall intensity, and soil type (or land use type). Many studies calculate runoff using the 'SCS (Soil Conservation Service) model' [19–21], which relies on a 'curve number' (CN) associated with different land use or land cover types. For example, the USDA (United States Department of Agriculture) Natural Resources Conservation Service divides soils into four hydrologic soil groups (e.g., low or high runoff potential), and each has a different range of CN.

Alternatively, some studies [22,23] use a runoff coefficient, which is a number relating the amount of runoff to the amount of precipitation received and is empirically derived depending on different land use or land cover types. The classification of land use and land cover types depends on the literature and will be discussed further under 'land use' data in Section 4.3. Surface flow captures the hydrological processes of redistributing rainwater, e.g., flow direction and velocity. Digital elevation models (DEM) or geographic information systems (GIS) are commonly used in the modelling (e.g., [18,19]), since surface flow is typically influenced by land elevation, gravity, and natural water bodies.

Drainage flow captures the waterflow through the artificial urban drainage network. Calculation of drainage flow focuses on the flow rate and/or discharge capacity and often uses parameters, such as manhole locations, pipe dimensions, and coverage area.

The calibration and validation of hydrological models plays an essential role in simulation and prediction, which concerns evaluating and/or adjusting model performance, or addressing model uncertainty [24]. In the first case, simulation or prediction results (i.e., model outputs) are compared against ground truth data that are collected from real flooding incidents. These may include water depth and flooded areas [25,26] as well as the volume of waterflow through the drainage system [27]. In the second case, the focus is on evaluating and minimizing variations in model outputs when model parameters are varied [18,28,29]. It is widely acknowledged that studies in this area are still lacking [24,26,28,30], primarily due to insufficient data being collected from flooding events [24,28].

### 3.2. Risk Analysis

This task focuses on quantifying the vulnerability and potential damage of flooding using different metrics, such as those for buildings and traffic, sometimes in monetary terms. Related studies are often based on results from *inundation simulation and prediction* (in the following, we use the italic font style to indicate a 'task'). Shi et al. mapped the simulation results to locations of 'old-style' buildings to calculate 'waterlogging exposure index' (WEI) that depends on the area size of such buildings [31]. Su et al. studied the impact of urban flooding on traffic, by combining simulated water depth and other factors, such as driver behaviors and vehicle speed during rainstorms, to calculate road congestion [32].

Another similar study on traffic is Han et al. [33]. Quan used predicted flood depth to calculate building structure and content damage [22]. As such, flooding risks are calculated in terms of financial cost in replacing the damaged buildings and contents. Buildings and contents are often classified into different bands in terms of the rebuild/replacement cost, while damage is calculated based on the water depth and inundation duration. Ferligoj evaluated the financial cost based on predicted urban flooding severity, and they considered a wide range of factors, such as building type, structural damage to buildings, utility infrastructure, rebuild/cleaning cost, public facility, and traffic disruption [34].

Without using simulation, Lin et al. calculated a risk index for different catchments based on the population and public facilities (education and medical related) and then mapped this index to historical data on flooding spots in the city [35]. Both parts of the data are combined to discuss the city's vulnerability to flood. Wu et al. followed a similar approach but based its calculation on simulation results [36].

### 3.3. Flood Monitoring

This task focuses on capturing the real-time progression of urban flooding incidents, based on various data sources, such as IoT (Internet of Things) sensors, aerial images, and social media. IoT sensors cover a wide range of devices, such as rainfall sensors, water level gauges, and CCTV (Closed-Circuit TeleVision) footage. Monitoring systems were described as involving a network of WiFi and RFID enabled IoT devices to track real-time data of rainfall, drainage discharge, and flood depth [37–39]. Jiang et al. used CCTV images of flooded streets to calculate the water depth with respect to reference objects, such as lamp posts and post boxes [40]. She et al. used GPS trajectory data collected from taxis to identify flooded streets in real time [41]. The idea is that such data indicate the formation of congestion, which, during heavy rainfall, may be due to flooding.

Aerial images (in the following, we use underline font style to indicate a 'data source') of a flooded area can also be used to identify flooding incidents and severity. Such images can be taken by Unmanned Aerial Vehicles (UAV) [42,43], or satellites [44]. However, the satellite image quality is sensitive to weather. The use of social media has been on the rise, such as Tweeter [45], Weibo [46,47]. For example, Rosser et al. used image data from Flickr and satellite images to estimate the water depth during urban flooding incidents [48]. Choi and Bae monitored tweets during heavy rainfall to identify flooding spots [45].

### 3.4. Response and Evacuation Planning

This task aims to analyze and optimize emergency plans, such as logistics allocation and routing. This is a complex task as it concerns different disciplines, such as transport planning, engineering, demographics, and human psychology and behaviors.

Related studies are often based on a simplified scenario. For example, Chang et al. [49] and Chen et al. [50] studied the optimization of emergency plans based on a wide range of factors, such as rescue resources (e.g., vehicles, labor, and rescue centers), their capacity and cost, geospatial distribution and coverage, road networks, and organizational structures of emergency response agencies. This could inform the setup of rescue bases during the disaster and the quantity of rescue equipment required. Simonovic and Ahmad studied the acceptance of evacuation orders by residents and combined this information with population demographics to predict the number of evacuees and the time required for them to reach safety [51]. Ferligoj studied how the emergency and evacuation resources potentially address these risks of a city, such as the fire station coverage and public facilities that can be potentially used as evacuation centers [34].

### 3.5. Trend Analysis

This task examines the long-term temporal and geospatial patterns of flooding in an urban area, which can help inform water resource management research and policies. For example, Huang et al. studied inundation data for the Australian Murray–Darling Basin over a period of 10 years, to identify patterns, such as the inundation severity, frequency, and probability from both temporal and geospatial perspectives [52].

### 3.6. Cause Analysis

This task focuses on comparing or evaluating the contributing or mitigating factors to urban flooding. This can be done either quantitatively or qualitatively. Quantitative analysis often involves studying the correlation between flooding metrics (e.g., the extent, severity, and frequency) and data related to the contributing factors over a long period of time. For example, Wu et al. found that the change in land use and delayed drainage network development had a larger impact than precipitation on flooding frequency across a period of over 50 years [23]. Yu and Coulthard quantified the effect of different factors from the perspective of mitigation, e.g., to what extent a simulated increase in drainage network coverage can reduce flooding [18].

Qualitative studies usually discuss urban flooding causes from a theoretical point of view [53–56] without data collection or analysis, they or study public perceptions through questionnaire and focus group analysis [57].

### 3.7. Conceptual Modelling

This task involves addressing conceptual models and frameworks for information systems of urban flooding emergency management. This is also an important task because it focuses on data representation and data/system integration and enables data compatibility and system interoperability in developing information systems for urban flooding management. Studies in this area typically describe data models for urban flooding management based on ontological models [5,6,8]. However, empirical work that utilizes such data models or justifies their benefits in practice is still lacking.

### 3.8. Policy Analysis

This task aims to explain, evaluate, and reflect on government initiatives, planning documents and policies around urban flooding. Researchers use quantitative metrics for evaluation [58,59]. For example, Han et al. proposed 20 indicators for quantifying urban flood management plans based on the 5R framework for flood risk management [58]. Some studies on *policy analysis* do not make use of existing data but produce data potentially useful for other tasks, e.g., public perceptions/awareness [57] and human behavior.

## 4. Data Used in Urban Flooding Management

The data are categorized based on the subject nature as hydrological data, topographic data, urban planning data, traffic data, disaster damage data, census data, human perception and behavior data, and parameter data. Such data are collected through various sources, including curated sources, aerial images, radar images, physical sensors, social media, open web datasets, web news (excluding social media), and surveys and interviews. Data of the same category can be acquired from multiple sources.

### 4.1. Hydrological Data

*Hydrological data* often include data related to precipitation and watercourse evaporation in the context of urban flooding. Precipitation or rainfall is typically measured by intensity and duration. For intensity, most estimates an average from historical data, and the timespan for consideration can be annual [23], daily [23], or hourly [15]. For duration, minute-based [16,17], hourly [21,31,32,60–62], daily [18,33,63], and multi-day-based [23] models are all reported. It is also noticed all studies made use of historical data, and real-time precipitation are only addressed in theoretical studies on data and system integration models [5,8,38].

Watercourse data describe the watercourse network [23,31] or dynamic data about the water flows [33,52]. It is used in those studies where the areas of concern contain significant water bodies, such as rivers, canals, and lakes. Evaporation data describe the speed of which surface water vapors. It is used in only a handful of studies [18,32], since evaporation is recognized to play a minor role. *Hydrological data* are predominantly collected from curated sources, particularly those managed by administrative bodies (e.g., weather stations, research institutions). Evaporation rates are simply treated as constant variables based on scientific parameter data.

### 4.2. Topographic Data

*Topographic data* in the context of urban flooding predominantly refer to data recording the elevation and form of terrain that is divided into units of areas. The majority simply dividing a geographical area into equal sized grids, and the granularity of the grids varies, ranging from several square meters to hundreds of square meters to square kilometers.

*Topographic data* are widely used in *inundation simulation and prediction* (e.g., [17,64]) and *risk analysis* [32,36]. They are typically extracted from curated sources, such as Digital Elevation Models (DEM) built on various data inputs (e.g., high-resolution aerial images), and contour maps. *Topographical data* may come from authoritative organizations and government agencies [63,65] and contour maps. A couple of studies used open web datasets, such as OpenStreetMap [15].

### 4.3. Urban Planning Data

*Urban planning data* refer to a broad range of data describing the development and design of land use and the built environment in a human populated area. They are often coupled with *topographic data* [19,61]. Popularly-used *urban planning data* are summarized as below.

Drainage network data describe the layout of the drainage/sewage system in a city, and are mainly used to calculate discharge capacity and surface water flows. While some use specific parameters (the locations of manholes, pipe length, junction depth, conduit size, diameters, and pipe materials) to calculate the discharge at different locations [19,25,33,60], the majority use measurements from design standards [17–19,21,22]. Drainage network data are predominantly collected from curated sources.

Drainage monitoring data refer to data about discharge flow within the drainage network, often collected in real-time. These can be used for *flood monitoring* [38]. Such data needs to be collected through physical sensors.

Catchment areas data describe how an area is subdivided into smaller units, which are arguably not directly encoded in any sources but defined on an ad hoc basis. A common

approach is to define granular areas into equal sized shapes, such as the grid/cell/block system [15,18] or based on locations of manholes [19]. Huang et al. created areas of irregular shapes and sizes based on ecological and hydrological rules [52]. Luan et al. [25] used ArcGIS to digitize the properties of the confluence nodes of the drainage pipeline network in order to identify the flooded locations.

Land use determines the capacity of draining excess surface water by natural means, such as infiltration, and are primarily used for *cause analysis* [23], *inundation simulation and prediction* [18,66], and *risk analysis* [22]. Land use definitions are often ad hoc and case-driven. For example, Wu et al. [44] identified three types: agricultural, residential and industrial, and transport; Hou and Du [65] highlighted water body, green land and unused land; Yu and Coulthard [18] only distinguished urban from rural land; Hu et al. [16] defined six types: open land, low-density residence, green/garden area, high density residence, road, and lake. Land use data are primarily used for *cause analysis* [23], *inundation simulation and prediction* [18,66], and *risk analysis* [22], and are typically gathered from <u>curated sources</u>, such as administrative bodies, and can be analyzed based on satellite images [17,23] or radar images [67].

Point-of-Interest (POI) data describe public facilities, carrying information about their different degrees of attracting the crowd [35]. Zhang et al. hypothesized that different types of POIs (e.g., green area vs. stadiums) may be useful indications of land use and therefore can inform *risk analysis* [15]. Ferligoj identified not only common POIs (e.g., schools) but also those that may affect evacuation planning (e.g., hospitals) [34]. POI data can be collected from <u>open web datasets</u> [15,35] or <u>curated sources</u> [34].

Road network and public transport are both data related to the transportation system, and are widely used in *inundation simulation and prediction* [15], *risk analysis* [32], *flood monitoring* [41], and *response and evacuation planning* [34].

*4.4. Traffic Data*

*Traffic data* describe the movement of transportations in a human populated area. They record information, such as the volume, speed, direction, and location of traffic. In theory, they are particularly useful for *risk analysis* [32], *flood monitoring* [41], and response and evacuation modelling; however, they are rarely used. She et al. used GPS data uploaded by taxis to estimate traffic flows during rainstorms and predicted flooded streets based on the changes in traffic movement [41]. Su et al. used a traffic simulation model that takes input of a series of parameters, such as volume, speed, and traffic signal operation data [32]. *Traffic data* can be collected via <u>physical sensors</u> (i.e., GPS) [41] or <u>curated sources</u> [32].

*4.5. Disaster Damage Data*

*Disaster damage data* describe the extent of physical damage caused by urban flooding, and the economical and societal loss. The extent of physical damage is often described in terms of flooded areas and severity. These usually record the exact locations (e.g., streets, buildings, or as precise as geo-coordinates), and parameters, such as the area size, water depth, and duration. Such data can be obtained by analyzing textual and imagery data or geo-coordinate data in social media posts, and the analysis often involves image recognition, text analysis, or manual processing. Such data are often collected for *flood monitoring* [39] and are used in a wide range of tasks, including in *inundation simulation and prediction* [15], *cause analysis* [23], *risk analysis* [22], *response and evacuation planning* [49,51,68], and *trend analysis* [52].

Data for assessing economic and societal loss are less. Chang and Huang proposed an integrated ecological and economic system to evaluate the 'emergy' values of vulnerability [69]. Quan reported unitary costs (CNY/m$^2$) for replacing certain residence building structures [22]; while Han et al. [33] related different levels of water depth to traffic conditions measured by vehicle discharge per hour.

Damage data can be sourced from a wide range of channels. In addition to <u>curated sources</u> typically maintained by government administrative bodies [8,18,23,48,69], there is also

wide use of aerial images from satellites [44,48,52] and UAVs [42,43], radar images [67], physical sensors [37,40], social media [15,36,45–48], and web news [35,36,64].

### 4.6. Census Data

*Census data* describe the population of an administrative area and may include (but are not limited to) the size and density of a population, demographics, social economic status, and household composition. *Census data* are often needed to quantify vulnerability of an area during urban flooding in *risk analysis*, to inform *response and evacuation planning*, or to evaluate the damage. For example, Ferligoj used the population density of Buenos Aires to quantify access to public facilities (e.g., public transport and hospitals) [34]. Similar work can be found in [35,36].

*Census data* are predominantly collected from curated sources, typically government administrative data, such as China City Statistics Yearbook [70]. Some of these have been made available as open web datasets (e.g., the UK open census data).

### 4.7. Human Perception and Behaviour Data

*Human perception and behavior data* describe people's perceptions about urban flooding issues and understandings of how they behave during flooding incidents. Such data can benefit various tasks, such as *policy analysis* and *cause analysis* [57], and *response and evacuation planning* [51].

*Human perception and behavior data* are difficult to observe directly [71] and can be collected through surveys and interviews [51]. Social media also provides information on emotions, thoughts, and behaviors [45,48].

### 4.8. Parameter Data

*Parameter data* are those acting as configuration variables that are internal to a model, and are often found as arbitrary, ad hoc parameters in computational models or decision analysis models. For example, Chang et al. used parameters, such as equipment type, unit rent, average operating cost, and the unit penalty for shortage, in evaluating flood emergency plans [49].

Chen et al. evaluated evacuation plans by simulation, in which vehicles (e.g., ambulance and emergency communication vehicles) were assigned different degrees of mobility in terms of the number of grids they move at each single turn [50]. Concerning evacuation planning, Ding et al. defined the costs of different sizes of rescue team based on the labor cost, equipment rental cost, and material consumption [72]. Earlier in Section 3.1, some studies used the runoff coefficient as a constant parameter in their inundation simulation models.

The parameter values are typically estimated by considering scenarios that represent the possible realistic situations or learned from the statistics [49,72].

## 5. Discussion

### 5.1. Task Diversity and Data Heterogeneity

Table 1 shows the tasks and data involved in urban flooding management, and their links. Each type of task receives different levels of attention from research. The mostly studied tasks are *inundation simulation and prediction*, *risk analysis*, and *cause analysis*, which may be attributed to the fact that they can inform other tasks, such as *response and evacuation planning*. This also suggests the tasks in the mitigation, preparedness, and response phase have received much attention, while those in the recovery phase are overlooked.

**Table 1.** Tasks, data, and their links.

| | Inundation Simulation and Prediction | Risk Analysis | Flood Monitoring | Response and Evacuation | Trend Analysis | Cause Analysis | Conceptual Modelling | Policy Analysis |
|---|---|---|---|---|---|---|---|---|
| *Hydrological data* | [15–21,26–29,61,63–65,73,74] | [22,25,31–33,36,60,62,69,75] | [38,53] | [15,49,50,76] | [52,70] | [23] | [5,8,9,77,78] | [55,56,79] |
| *Topographic data* | [15–21,26,27,29,61,63–65] | [22,32,36,60,62,69,75,80–82] | [48,77] | [15,76] | | [83] | [5,8,9,77,78] | [56] |
| *Urban planning data* | [15–21,25–29,61,63–66,73,84] | [22,25,31–36,60,69,80,81] | [38,41,77] | [15,34,49,50,68,76] | [52,70] | [23,83] | [5,8,77,78] | [55,56,79] |
| *Traffic data* | | [32] | [41] | | | [83] | [8] | |
| *Disaster damage data* | [15,18,26,63,64] | [22,25,33,35,36] | [37,39,40,42–44,47,48] | [18,19,49–52,68,85] | [52] | [23,83,86] | [8] | |
| *Census data* | | [35,36,75,80] | | [34] | [70] | | [8] | |
| *Human perception and behavior data* | [63] | [32] | | | [71] | [57] | | [57] |
| *Parameter data* | [17–19,21,27–29,61] | | | [49–51,72,76] | | | | [58,59] |

The heterogeneity of data and the varying popularity levels of the studies on the data categories can also be observed. Clearly, this is partially attributed to the popularity of tasks, as well as the scope of these categories. For example, *urban planning data, hydrological data,* and *topographic data* are the most frequently used, because they are widely used in *inundation simulation and prediction* and each encompasses a wide range of data. IoT technology and social media has come to play a major role in urban flooding, which provides not only real-time information on flooding incidents [40] but also information on emotions, thoughts, discussions, and behaviors [45,48].

Heterogeneity can also be observed in terms of the different granularity of data used in different studies. For example, while a large number studies used precipitation data [15–18,21,23,31–33,63], the timespan at which such data are collected varies from minute-based to annually. *Topographic data*, such as elevation and form of terrain, are recorded for unit areas of different sizes, ranging from several square meters to square kilometers. The same is found for defining urban catchment areas [15,18,19,52].

When calculating drainage network capacity, some adopted design standards [17–21], others measured specific parameters (e.g., pipe length and manhole locations) to calculate more precise figures. The diversity of tasks and heterogeneity of data makes it difficult to compare and implement methods or systems. Some studies have proposed conceptual models and frameworks for integrating data and/or systems for urban flooding management [5,6,8]. However, they have been restricted to theoretical level, and implementation using such frameworks is lacking.

*5.2. Task-Data Links*

As shown in Table 1, the task–data links are a complex many-to-many relationship in the sense that one particular data category supports multiple tasks, while one particular task uses data from multiple categories.

There exists a lack of consensus in terms of their methodological approach, and the use of data categories in a task. For example, in *inundation simulation and prediction*, the existing methods vary in terms of the factors included in the model and their measurement. Chen et al. [17] and Meng et al. [19] had different calculations although they both included all three groups of factors. Rainfall data are often measured in different granularities, and units of areas for study are defined at different granularity levels. Different formalizations of land uses are also presented. As far as model calibration and validation is concerned, we notice a degree of inconsistency among the studies under '*inundation simulation and prediction*'.

Studies, such as [16,21], did not adopt calibration or validation; many based on subjective verification against historical field observations and reports [17,61,63–65] (e.g., simply checking if a predicted catchment matches or overlaps historical events); and only a

few [18,25,26,28,29] strictly followed standard measures and metrics. For the use of data categories, the studies used different data categories even if they addressed the same task. One of the most prominent examples is arguably *risk analysis*, in which the usage of all data categories has been reported while not every single study used them all.

While the 'inconsistency' partially reflects the innovative nature of research, it makes comparison and evaluation difficult. For example, there is a lack of understanding of the impacts that different data categories, or the granularity of data points and/or outputs have on simulation accuracy. Where the data uses are very different among different studies, such as for *flood monitoring*, it is unclear to what extent these different data categories can complement or replace each other. This situation makes it particularly challenging for researchers interested in gaining a thorough overview of this domain, and therefore, we suggest that future research looks more into comparative or evaluative studies.

*5.3. Research Opportunities*

5.3.1. Towards Data Integration and Openness

The issues brought by such levels of heterogeneity advocate for data integration and standardization. The benefits could be manifold, with the primary being facilitating wider access to data for better decision making, and enabling 'system integration' [8], which is seen as the ultimate trend for urban flooding research. In a similar direction, the case-based reasoning (CBR) approach has been studied in emergency responses in general [7], which retrieve similar cases to inform decision making during urban flooding disasters. Ontology is often used to model the domain knowledge (e.g., actions and consequences) and index historical cases, and provides a means for data integration and standardization.

Ontology has been proposed for integrating data from heterogeneous sources for urban flooding management [8]; however, it is unclear how this can be used in the practice or the benefits. Reflecting from our literature review, one reason is its highly interdisciplinary nature. As previously discussed, data may be acquired from different sources, indexed using different formats and standards, and possessed by different owners, which makes data integration in this domain particularly challenging and requires an orchestrated effort of many parties.

However, we recognize that achieving data standardization and integration is a long-term goal that requires orchestrated effort from both researchers and practitioners. We therefore also argue that the first step towards achieving this is enabling 'data openness'. Future research should endeavor to use non-proprietary data, released under open licenses and in a format that supports data sharing and reusability, such as following the linked data practice. This will encourage people to use existing datasets, thus, reducing the data heterogeneity at different levels. Section 4 highlighted particular datasets that are available as open web datasets. However, these are far from sufficient, and promoting data openness still deserves significant effort.

5.3.2. Recovery-Related Topics

The recovery phase has received less attention compared with the other three. Recovery mainly includes the relief and reduction of current and further losses in the aftermath of a disaster. The short-term recovery may overlap with the response phase, which involves specific tasks, such as providing essential public health and safety services, restoring interrupted utility and other essential services, reestablishing transportation routes, and providing food and shelter for those displaced by the incident.

The long-term recovery may involve some of the same tasks but may continue for a longer term, depending on the severity and extent of the damage sustained. For information system, the recovery phase is often a focus for post-disaster assessment [2], in which models are employed to generate products for evaluating the destruction from the urban flooding disaster (e.g., infrastructure damage, life and economic losses, etc.). Other topics may also include supplies scheduling. Future research is expected to address these topics, particularly to capitalize on 'big data' and 'IoT' opportunities.

### 5.3.3. Exploring New Data

New types of data have come to play an important role in urban flooding management and will drive innovation in both research and practice. For example, the use of *urban planning data*, such as POI, road network data, and social media data in *risk analysis* is increasing. These data are not fully explored yet. For example, the data from physical sensors and social media can be analyzed for real-time decision making in *response and evacuation planning*, not just for *flood monitoring* and post hoc analysis. With the real-time data, we can better answer questions, including "Does the real-time inundation deviate from the prediction? If yes, how should the evacuation plan respond to the deviations?".

In addition, existing studies generally use data from an extremely limited number of sources or devices, and the future research is expected to make extensive use of data from multiple sources or devices so as to obtain information as well as mutual verification.

### 5.3.4. Social Factors

There is a need to considerate social factors, particularly for the tasks where humans play an important role. Studies have shown how human perception and behavioral data can affect *risk analysis* and inform *response and evacuation planning* [32,51]. Nevertheless, more work needs to be done to establish what 'social' factors (e.g., rush hour commuting patterns, population demographics, POI, and public gathering events) should be considered and how these can be captured in data.

### *5.4. Limitations*

This study could have been enriched from several angles. First, the task–data links could have been discussed in terms of the methodologies and (where appropriate) information systems used. Second, commonly-used datasets for different data categories could have been analyzed. These options are not pursued due to the space limit but also due to the highly heterogeneous nature of urban flooding data sources in the literature. For example, while some clearly describe the algorithms and systems used for *inundation simulation and prediction* (e.g., [64]), others do not (e.g., [17,20]).

There is also a limitation concerning the literature. We did not cover the entire literature in this field but rather those studies with an empirical focus on what data and tasks are being used. We also observe that a significant portion of our selected literature used case studies from China or were written by scholars affiliated with Chinese institutions. Although this may appear to be a bias, we argue that this reflects the nature of this research topic, and the recent trends in this area of research. Urban flooding often happens due to the urban infrastructure development failing to meet the pressure from rapid urbanization and economic development, which is more typically seen in developing countries.

Our literature search strategy did not intentionally exclude literature from other areas of the world but only focused on more recent studies. For these reasons, some relevant studies could have been missed. However, this degree of 'incompleteness' does not undermine the value of our work, as we noticed that our literature search converged to the task types and data categories identified above. Increasing the literature coverage would be more important for other purposes, such as to gain a thorough understanding of the methods, systems, and specific datasets.

## 6. Conclusions and Recommendations

Urban flooding has become one of the most frequent and serious natural hazards. Managing urban flooding is inherently complex due to its highly interdisciplinary nature, the task diversity and data heterogeneity. This is reflected by the extent of heterogeneity in the literature, where we notice a lack of standardization and consistency in terms of the tasks studied, the data used, or the approaches adopted. Inevitably, this makes it difficult for researchers to gain a thorough understanding of the domain or to compare existing studies.

As such, this study set out to understand how heterogenous data are used to tackle urban flooding. From the selected 69 articles on this topic, eight categories of tasks and eight categories of data were defined, and the links between tasks and data were identified by synthesizing what data were used to support the tasks in the studies. While the majority of the literature focused on case studies from developing countries—particularly China—as we discussed before, we believe this reflects the nature of the problem being closely related to economic development and urbanization.

It is worth noting that the practical approaches adopted across the globe are largely indifferent, including those from Europe [18,49], Australia [45], and America [17,35]. In fact, the approaches adopted in the current literature are very much based on the earlier ones developed by European and US authors, such as 'Mike Urban' used in simulation [25,68]. In other regions of the world, we note a few government-led initiatives. The EU Floods Directive adopted in 2007 aims to coordinate flood prevention, protection, and preparedness within and between EU member states.

It has been acknowledged that the Flood Directive plays a positive role in the standardization of flood risk assessment and management [87]. Member states have begun implementation of Flood Risk Management Plans; however, improvements are still needed. In the US, Government Accountability Office (GAO) created the Disaster Resilience Framework to guide federal efforts to develop disaster resilience. Resilience is a concept that attracted attention from the research community in recent years, which refers to the ability to adapt to and recover from hazards, shocks, or stresses without compromising long-term prospects for development.

The Disaster Resilience Framework lays out three broad principles, including (1) information, which is about giving federal and nonfederal decision makers authoritative and understandable information to help identify current and future risks, as well as the impact of risk-reduction strategies; (2) integration, which is about enabling decision makers to take coherent and coordinated actions; and (3) incentives, which is about making long-term, forward-looking, risk-reduction investments more viable and attractive among competing priorities [88].

Nevertheless, it is recognized that, despite these efforts, more work remains to be completed around further integration and standardization [87], and this may require orchestrated efforts from researchers and practitioners across the globe. Reflecting on this study, we summarize the following recommendations for future research and practice in tackling urban flooding. First, future implementations of big data-driven solutions can be built on data integration frameworks, particularly those driven by ontology-based technologies for enabling CBR-based approaches. This paper lays the foundation for such studies aiming to develop a data-driven approach to tackle urban waterlogging.

Second, recovery- and resilience-related topics are expected to be further explored, particularly to capitalize on the big data opportunities. Recovery and resilience encompass more than the restoration of a community's physical structures to pre-disaster conditions. The ability of a community to accelerate the recovery process begins with its efforts in pre-disaster preparedness, including coordinating with whole community partners, mitigating risks, incorporating continuity planning, identifying resources, and developing capacity to effectively manage the recovery process as well as through collaborative and inclusive planning processes. Due to big data, we will be able to observe the dynamic recovery trajectories and to better estimate and predict the resilience of communities.

Third, we recommend the use of IoT and social media data to enable dynamic monitoring and response in urban waterlogging management. It is important to note that social media data contain rich information about human behaviors and social factors, which is necessary to be considered when developing technical solutions.

Last, when it comes to the heterogeneity of data in this field, we call for the adoption of open data standards to reduce data heterogeneity at different levels and facilitate data sharing and reuse in this field. We also suggest that comparative and evaluative studies

be undertaken to understand the impact of different data categories and analytics for a particular task.

**Author Contributions:** Conceptualization, Z.Z. and M.R.; methodology, J.Z. and M.R.; literature review, M.R., Z.Z. and J.Z.; summarization, Z.Z.; writing—original draft preparation, M.R., Z.Z. and J.Z.; writing—review and editing, M.R. and L.M.; funding acquisition, M.R. and Z.Z. All authors have read and agreed to the published version of the manuscript.

**Funding:** This research was partly funded by Global Challenges Research Fund (GCRF) quality-related research (QR), and the Renmin University of China. The APC was funded by the University of Sheffield Institutional Open Access Fund provided through UK Research and Innovation.

**Institutional Review Board Statement:** Not applicable.

**Informed Consent Statement:** Not applicable.

**Data Availability Statement:** No new data were created or analyzed in this study. Data sharing is not applicable to this article.

**Acknowledgments:** For the purpose of open access, the author has applied a Creative Commons Attribution (CC BY) license to any Author Accepted Manuscript version arising.

**Conflicts of Interest:** The authors declare no conflict of interest.

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
