# Peer review of "Understanding the Use of Heterogenous Data in Tackling Urban Flooding: An Integrative Literature Review"

_water, doi:10.3390/w14142160_

Round 1
Reviewer 1 Report
I like this kind of integrative review very much. It represents a good source of information for other researchers. It gains more citation and therefore helps the journal to gain a higher impact.
The article is worth publishing. I have only minor comments:
1- I did not recognize solid conclusions. Please write the recommendations in a separate section
7. Recommendations
2- I suggest the use of the information of the methodology of Ref. 14 and your approach to creating a detailed flowchart to make it easy to follow the adopted methodology by others.
3- Line 395: remove the extra dot.
Check all occurrences (heading of sections 3.8, 5.4, ..etc)
Reviewer 2 Report
General Comments
The data used to manage urban floods must inherently be heterogeneous. This is due to the complexity of the problem of urban flood management faced by emergency services in every city and in every country. As noted by the authors of the article, there are no established global standards in the approach to urban flood management. Each country and even every major city has a different way of managing floods and uses different data, mainly those that are easily available to emergency services. Only the European Union has created a kind of standard for dealing with floods, mainly fluvial ones, by developing the EU Flood Directive, which all EU countries should adopt and implement its recommendations, although this has not entirely happened. The authors did the research on the basis of 66 articles mainly by authors of Chinese nationality and presented in my opinion only those studies and experiences that come from Chinese cities. Maybe I'm wrong. I am not able to read all the articles taken for research. However, when looking at urban flood management holistically, it would also be worth taking into account the European approach (EU Flood Directive) and the experience of researchers from the US. Nevertheless, I consider the reviewed article to be a valuable attempt to grasp the problems of urban flood management resulting from the multitude and diversity of data and the complexity of the tasks that must be taken into account in its implementation. That is why I am in favor of publishing the reviewed article.
Minor remarks
- I propose to change the term “waterlogging” to “urban floods” throughout the article. This term is commonly used.
- I propose to consider using the term “recovery” and use instead the term “resilience”, which in the literature on the topic of the article has broader connotations of flooding the city.
- Line 355: please correct "CNY/m2" to CNY/m2.
- Line 395: please correct ". Task" to Task.
- Line 499: please correct ". Limitations" to Limitations.
